# IntroVAE: Introspective Variational Autoencoders for Photographic Image Synthesis

**Huaibo Huang, Zhihang Li, Ran He**[*] **Zhenan Sun, Tieniu Tan**

[1]School of Artificial Intelligence, University of Chinese Academy of Sciences, Beijing, China
[2]Center for Research on Intelligent Perception and Computing, CASIA, Beijing, China
[3]National Laboratory of Pattern Recognition, CASIA, Beijing, China
[4]Center for Excellence in Brain Science and Intelligence Technology, CAS, Beijing, China
`huaibo.huang@cripac.ia.ac.cn`
`{zhihang.li, rhe, znsun, tnt}@nlpr.ia.ac.cn`

## Abstract

We present a novel introspective variational autoencoder (IntroVAE) model for synthesizing high-resolution photographic images. IntroVAE is capable of self-evaluating the quality of its generated samples and improving itself accordingly. Its inference and generator models are jointly trained in an introspective way. On one hand, the generator is required to reconstruct the input images from the noisy outputs of the inference model as normal VAEs. On the other hand, the inference model is encouraged to classify between the generated and real samples while the generator tries to fool it as GANs. These two famous generative frameworks are integrated in a simple yet efficient single-stream architecture that can be trained in a single stage. IntroVAE preserves the advantages of VAEs, such as stable training and nice latent manifold. Unlike most other hybrid models of VAEs and GANs, IntroVAE requires no extra discriminators, because the inference model itself serves as a discriminator to distinguish between the generated and real samples. Experiments demonstrate that our method produces high-resolution photo-realistic images (e.g., CELEBA images at $1024^2$), which are comparable to or better than the state-of-the-art GANs.

## 1 Introduction

In the recent years, many types of generative models such as autoregressive models [38, 37], variational autoencoders (VAEs) [20, 32], generative adversarial networks (GANs) [13], real-valued non-volume preserving (real NVP) transformations [7] and generative moment matching networks (GMMNs) [24] have been proposed and widely studied. They have achieved remarkable success in various tasks, such as unconditional or conditional image synthesis [22, 27], image-to-image translation [25, 46], image restoration [5, 17] and speech synthesis [12]. While each model has its own significant strengths and limitations, the two most prominent models are VAEs and GANs. VAEs are theoretically elegant and easy to train. They have nice manifold representations but produce very blurry images that lack details. GANs usually generate much sharper images but face challenges in training stability and sampling diversity, especially when synthesizing high-resolution images.

Many techniques have been developed to address these challenges. LAPGAN [6] and StackGAN [41] train a stack of GANs within a Laplacian pyramid to generate high-resolution images in a coarse-to-fine manner. StackGAN-v2 [42] and HDGAN [43] adopt multi-scale discriminators in a tree-like structure. Some studies [11, 39] have trained a single generator with multiple discriminators

---

[*]Ran He is the corresponding author.

to improve the image quality. PGGAN [18] achieves the state-of-the-art by training symmetric generators and discriminators progressively. As illustrated in Fig. 1(a) (A, B, C, and D show the above GANs respectively), most existing GANs require multi-scale discriminators to decompose high-resolution tasks to from-low-to-high resolution tasks, which increases the training complexity. In addition, much effort has been devoted to combining the strengths of VAEs and GANs via hybrid models. VAE/GAN [23] imposes a discriminator on the data space to improve the quality of the results generated by VAEs. AAE [28] discriminates in the latent space to match the posterior to the prior distribution. ALI [10] and BiGAN [8] discriminate jointly in the data and latent space, while VEEGAN [35] uses additional constraints in the latent space. However, hybrid models usually have more complex network architectures (as illustrated in Fig. 1(b), A, B, C, and D show the above hybrid models respectively) and still lag behind GANs in image quality [18].

To alleviate this problem, we introduce an introspective variational autoencoder (IntroVAE), a simple yet efficient approach to training VAEs for photographic image synthesis. One of the reasons why samples from VAEs tend to be blurry could be that the training principle makes VAEs assign a high probability to training points, which cannot ensure that blurry points are assigned to a low probability [14]. Motivated by this issue, we train VAEs in an introspective manner such that the model can self-estimate the differences between generated and real images. In the training phase, the inference model attempts to minimize the divergence of the approximate posterior with the prior for real data while maximize it for the generated samples; the generator model attempts to mislead the inference model by minimizing the divergence of the generated samples. The model acts like a standard VAE for real data and acts like a GAN when handling generated samples. Compared to most VAE and GAN hybrid models, our version requires no extra discriminators, which reduces the complexity of the model. Another advantage of the proposed method is that it can generate high-resolution realistic images through a single-stream network in a single stage. The divergence object is adversarially optimized along with the reconstruction error, which increases the difficulty of distinguishing between the generated and real images for the inference model, even for those with high-resolution. This arrangement greatly improves the stability of the adversarial training. The reason could be that the instability of GANs is often due to the fact that the discriminator distinguishes the generated images from the training images too easily [18, 30].

Our contribution is three-fold. i) We propose a new training technique for VAEs, that trains VAEs in an introspective manner such that the model itself estimates the differences between the generated and real images without extra discriminators. ii) We propose a single-stream single-stage adversarial model for high-resolution photographic image synthesis, which is, to our knowledge, the first feasible method for GANs to generate high-resolution images in such a simple yet efficient manner, e.g., CELEBA images at $1024^2$. iii) Experiments demonstrate that our method combines the strengths of GANs and VAEs, producing high-resolution photographic images comparable to those produced by the state-of-the-art GANs while preserving the advantages of VAEs, such as stable training and nice latent manifold.

## 2   Background

As our work is a specific hybrid model of VAEs and GANs, we start with a brief review of VAEs, GANs and their hybrid models.

**Variational Autoencoders (VAEs)** consist of two networks: a generative network (Generator) $p_\theta(x|z)$ that samples the visible variables $x$ given the latent variables $z$ and an approximate inference network (Encoder) $q_\phi(z|x)$ that maps the visible variables $x$ to the latent variables $z$ which approximate a prior $p(z)$. The object of VAEs is to maximize the variational lower bound (or evidence lower bound, ELBO) of $p_\theta(x)$:

$$log p_\theta(x) \geq E_{q_\phi(z|x)} \log p_\theta(x|z) - D_{KL}(q_\phi(z|x)||p(z)). \qquad (1)$$

The main limitation of VAEs is that the generated samples tend to be blurry, which is often attributed to the limited expressiveness of the inference models, the injected noise and imperfect element-wise criteria such as the squared error [23, 45]. Although recent studies [4, 9, 21, 34, 45] have greatly improved the predicted log-likelihood, they still face challenges in generating high-resolution images.

**Generative Adversarial Networks (GANs)** employ a two-player min-max game with two models: the generative model (Generator) G produces samples $G(z)$ from the prior $p(z)$ to confuse the

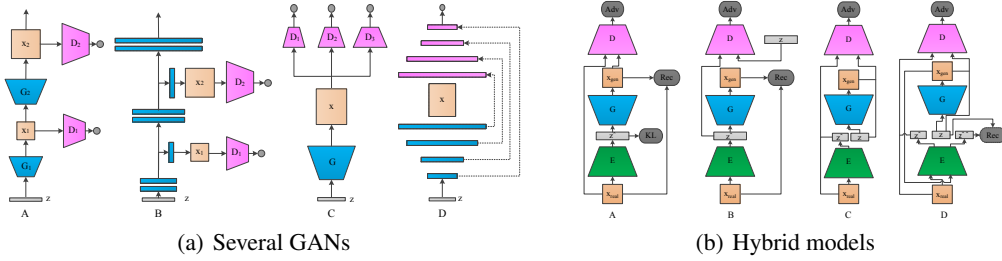

(a) Several GANs        (b) Hybrid models

Figure 1: Overviews of several typical GANs for high-resolution image generation and hybrid models of VAEs and GANs.

discriminator $D(x)$, while $D(x)$ is trained to distinguish between the generated samples and the given training data. The training object is

$$\min_{G} \max_{D} E_{x \sim p_{data}(x)}[\log D(x)] + E_{z \sim p_z(z)}[\log(1 - D(G(z)))]. \tag{2}$$

GANs are promising tools for generating sharp images, but they are difficult to train. The training process is usually unstable and is prone to mode collapse, especially when generating high-resolution images. Many methods [44, 1, 2, 15, 33] have been attempted to improve GANs in terms of training stability and sample variation. To synthesize high-resolution images, several studies have trained GANs in a Laplacian pyramid [6, 41] or a tree-like structure [42, 43] with multi-scale discriminators [11, 29, 39], mostly in a coarse-to-fine manner, including the state-of-the-art PGGAN [18].

**Hybrid Models of VAEs and GANs** usually consist of three components: an encoder and a decoder, as in autoencoders (AEs) or VAEs, to map between the latent space and the data space, and an extra discriminator to add an adversarial constraint into the latent space [28], data space [23], or their joint space [8, 10, 35]. Recently, Ulyanov et al. [36] propose adversarial generator-encoder networks (AGE) that shares some similarity with ours in the architecture of two components, while the two models differ in many ways, such as the design of the inference models, the training objects, and the divergence computations. Brock et al. [3] also propose an introspective adversarial network (IAN) that the encoder and discriminator share most of the layers except the last layer, and their adversarial loss is a variation of the standard GAN loss. In addition, existing hybrid models, including AGE and IAN, still lag far behind GANs in generating high-resolution images, which is one of the focuses of our method.

## 3 Approach

In this section, we train VAEs in an introspective manner such that the model can self-estimate the differences between the generated samples and the training data and then updates itself to produce more realistic samples. To achieve this goal, one part of the model needs to discriminate the generated samples from the training data, and another part should mislead the former part, analogous to the generator and discriminator in GANs. Specifically, we select the approximate inference model (or encoder) of VAEs as the discriminator of GANs and the generator model of VAEs as the generator of GANs. In addition to performing adversarial learning like GANs, the inference and generator models are also expected to train jointly for the given training data to preserve the advantages of VAEs.

There are two components in the ELBO objective of VAEs, a log-likelihood (autoencoding) term $L_{AE}$ and a prior regularization term $L_{REG}$, which are listed below in the negative version:

$$L_{AE} = -E_{q_\phi(z|x)} \log p_\theta(x|z), \tag{3}$$

$$L_{REG} = D_{KL}(q_\phi(z|x)||p(z)). \tag{4}$$

The first term $L_{AE}$ is the reconstruction error in a probabilistic autoencoder, and the second term $L_{REG}$ regularizes the encoder by encouraging the approximate posterior $q_\phi(z|x)$ to match the prior $p(z)$. In the following, we describe the proposed introspective VAE (IntroVAE) with the modified combination objective of these two terms.

## 3.1 Adversarial distribution matching

To match the distribution of the generated samples with the true distribution of the given training data, we use the regularization term $L_{REG}$ as the adversarial training cost function. The inference model is trained to minimize $L_{REG}$ to encourage the posterior $q_\phi(z|x)$ of the real data $x$ to match the prior $p(z)$, and simultaneously to maximize $L_{REG}$ to encourage the posterior $q_\phi(z|G(z'))$ of the generated samples $G(z')$ to deviate from the prior $p(z)$, where $z'$ is sampled from $p(z)$. Conversely, the generator $G$ is trained to produce samples $G(z')$ that have a small $L_{REG}$, such that the samples' posterior distribution approximately matches the prior distribution.

Given a data sample $x$ and a generated sample $G(z)$, we design two different losses, one to train the inference model $E$, and another to train the generator $G$:

$$L_E(x, z) = E(x) + [m - E(G(z))]^+,\tag{5}$$

$$L_G(z) = E(G(z)),\tag{6}$$

where $E(x) = D_{KL}(q_\phi(z|x)||p(z))$, $[\cdot]^+ = max(0, \cdot)$, and $m$ is a positive margin. The above two equations form a min-max game between the inference model $E$ and the generator $G$ when $E(G(z)) \leq m$, i.e., minimizing $L_G$ for the generator $G$ is equal to maximizing the second term of $L_E$ for the inference model $E$.*

Following the original GANs [14], we train the inference model $E$ to minimize the quantity $V(E, G) = \int_{x,z} L_E(x, z) p_{data}(x) p_z(z) dx dz$, and the generator $G$ to minimize the quantity $U(E, G) = \int_z L_G(z) p_z(z) dz$. In a non-parametric setting, i.e., $E$ and $G$ are assumed to have infinite capacity, the following theorem shows that when the system reaches a Nash equilibrium (a saddle point) $(E^*, G^*)$, the generator $G^*$ produces samples that are distinguishable from the given training distribution, i.e., $p_{G^*} = p_{data}$.

**Theorem 1.** Assuming that no region exists where $p_{data}(x) = 0$, $(E^*, G^*)$ forms a saddle point of the above system if and only if $(a)$ $p_{G^*} = p_{data}$ and $(b)$ $E^*(x) = \gamma$, where $\gamma \in [0, m]$ is a constant.
*Proof.* See Appendix A.

**Relationships with other GANs** To some degree, the proposed adversarial method appears to be similar to Energy-based GANs (EBGAN) [44], which views the discriminator as an energy function that assigns low energies to the regions of high data density and higher energies to the other regions. The proposed KL-divergence function can be considered as a specific type of energy function that is computed by the inference model instead of an extra auto-encoder discriminator [44]. The architecture of our system is simpler and the KL-divergence shows more promising properties than the reconstruction error [44], such as stable training for high-resolution images.

## 3.2 Introspective variational inference

As demonstrated in the previous subsection, playing a min-max game between the inference model $E$ and the generator $G$ is a promising method for the model to align the generated and true distributions and thus produce visual-realistic samples. However, training the model in this adversarial manner could still cause problems such as mode collapse and training instability, like in other GANs. As discussed above, we introduce IntroVAE to alleviate these problems by combining GANs with VAEs in an introspective manner.

The solution is surprisingly simple, and we only need to combine the adversarial object in Eq. (5) and Eq. (6) with the ELBO object of VAEs. The training objects for the inference model $E$ and the generator $G$ can be reformulated as below:

$$L_E(x, z) = E(x) + [m - E(G(z))]^+ + L_{AE}(x),\tag{7}$$

$$L_G(z) = E(G(z)) + L_{AE}(x).\tag{8}$$

The addition of the reconstruction error $L_{AE}$ builds a bridge between the inference model $E$ and the generator $G$ and results in a specific hybrid models of VAEs and GANs. For a data sample $x$ from the

training set, the object of the proposed method collapses to the standard ELBO object of VAEs, thus preserving the properties of VAEs; for a generated sample $G(z)$, this object generates a min-max game of GANs between $E$ and $G$ and makes $G(z)$ more realistic.

**Relationships with other hybrid models** Compared to other hybrid models [28, 23, 8, 10, 35] of VAEs and GANs, which always use a discriminator to regularize the latent code and generated data individually or jointly, the proposed method adds prior regularization into both the latent space and data space in an introspective manner. The first term in Eq. (7) (i.e., $L_{REG}$ in Eq. (4)) encourages the latent code of the training data to approximately follow the prior distribution. The adversarial part of Eq. (7) and Eq. (8) encourages the generated samples to have the same distribution as the training data. The inference model $E$ and the generator $G$ are trained both jointly and adversarially without extra discriminators.

Compared to AGE [36], the major differences are addressed in three-fold. 1) AGE is designed in an autoencoder-type where the encoder has one output variable and no noise term is injected when reconstructing the input data. The proposed method follows the original VAEs that the inference model has two output variables, i.e., $\mu$ and $\sigma$, to utilize the reparameterization trick, i.e., $z = \mu + \sigma \odot \epsilon$ where $\epsilon \sim N(0, I)$. 2) AGE uses different reconstruction errors to regularize the encoder and generator respectively, while the proposed method uses the reconstruction error $L_{AE}$ to regularize both the encoder and generator. 3) AGE computes the KL-divergence using batch-level statistics, i.e., $m_j$ and $s_j$ in Eq. (7) in [36], while we compute it using the two batch-independent outputs of the inference model, i.e., $\mu$ and $\sigma$ in Eq. (9). For high-resolution image synthesis, the training batch-size is usually limited to be very small, which may harm the performance of AGE but has little influence on ours. As AGE is trained on $64 \times 64$ images, we re-train AGE and find it hard to converge on $256 \times 256$ images; there is no improvement even when replacing AGE's network with ours.

### 3.3 Training IntroVAE networks

Following the original VAEs [20], we select the centered isotropic multivariate Gaussian $N(0, I)$ as the prior $p(z)$ over the latent variables. As illustrated in Fig. 2, the inference model $E$ is designed to output two individual variables, $\mu$ and $\sigma$, and thus the posterior $q_\phi(z|x) = N(z; \mu, \sigma^2)$. The input $z$ of the generator $G$ is sampled from $N(z; \mu, \sigma^2)$ using a reparameterization trick: $z = \mu + \sigma \odot \epsilon$ where $\epsilon \sim N(0, I)$. In this setting, the KL-divergence $L_{REG}$ (i.e., $E(x)$ in Eq. (7) and Eq. (8)), given $N$ data samples, can be computed as below:

$$L_{REG}(z; \mu, \sigma) = \frac{1}{2} \sum_{i=1}^{N} \sum_{j=1}^{M_z} (1 + \log(\sigma_{ij}^2) - \mu_{ij}^2 - \sigma_{ij}^2), \tag{9}$$

where $M_z$ is the dimension of the latent code $z$.

For the reconstruction error $L_{AE}$ in Eq. (7) and Eq. (8), we choose the commonly-used pixel-wise mean squared error (MSE) function. Let $x_r$ be the reconstruction sample, $L_{AE}$ is defined as:

$$L_{AE}(x, x_r) = \frac{1}{2} \sum_{i=1}^{N} \sum_{j=1}^{M_x} \|x_{r,ij} - x_{ij}\|_F^2, \tag{10}$$

where $M_x$ is the dimension of the data $x$.

Similar to VAE/GAN [23], we train IntroVAE to discriminate real samples from both the model samples and reconstructions. As shown in Fig. 2, these two types of samples are the reconstruction samples $x_r$ and the new samples $x_p$. When the KL-divergence object of VAEs is adequately optimized, the posterior $q_\phi(z|x)$ matches the prior $p(z)$ approximately and the samples are similar to each other. The combined use of samples from $p(z)$ and $q_\phi(z|x)$ is expected to provide a more useful signal for the model to learn more expressive latent code and synthesize more realistic samples. The total loss functions for $E$ and $G$ are respectively redefined as:

$$\begin{aligned} L_E &= L_{REG}(z) + \alpha \sum_{s=r,p} [m - L_{REG}(z_s)]^+ + \beta L_{AE}(x, x_r) \\ &= L_{REG}(Enc(x)) + \alpha \sum_{s=r,p} [m - L_{REG}(Enc(ng(x_s)))]^+ + \beta L_{AE}(x, x_r), \end{aligned} \tag{11}$$

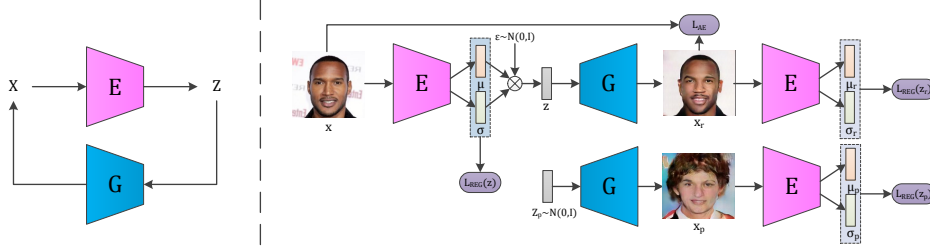

Figure 2: The architecture and training flow of IntroVAE. The left part shows that the model consists of two components, the inference model $E$ and the generator $G$, in a circulation loop. The right part is the unrolled training flow of the proposed method.

---

**Algorithm 1** Training IntroVAE model

1: $\theta_G, \phi_E \leftarrow$ Initialize network parameters
2: **while** not converged **do**
3: $\quad X \leftarrow$ Random mini-batch from dataset
4: $\quad Z \leftarrow Enc(X)$
5: $\quad Z_p \leftarrow$ Samples from prior $N(0, I)$
6: $\quad X_r \leftarrow Dec(Z), X_p \leftarrow Dec(Z_p)$
7: $\quad L_{AE} \leftarrow L_{AE}(X_r, X)$
8: $\quad Z_r \leftarrow Enc(ng(X_r)), Z_{pp} \leftarrow Enc(ng(X_p))$
9: $\quad L_{adv}^E \leftarrow L_{REG}(Z) + \alpha\{[m - L_{REG}(Z_r)]^+ + [m - L_{REG}(Z_{pp})]^+\}$
10: $\quad \phi_E \leftarrow \phi_E - \eta \nabla_{\phi_E}(L_{adv}^E + \beta L_{AE}) \qquad \qquad \triangleright$ Perform Adam updates for $\phi_E$
11: $\quad Z_r \leftarrow Enc(X_r), Z_{pp} \leftarrow Enc(X_p)$
12: $\quad L_{adv}^G \leftarrow \alpha\{L_{REG}(Z_r) + L_{REG}(Z_{pp})\}$
13: $\quad \theta_G \leftarrow \theta_G - \eta \nabla_{\theta_G}(L_{adv}^G + \beta L_{AE}) \qquad \qquad \triangleright$ Perform Adam updates for $\theta_G$
14: **end while**

---

$$L_G = \alpha \sum_{s=r,p} L_{REG}(Enc(x_s)) + \beta L_{AE}(x, x_r), \qquad (12)$$

where $ng(\cdot)$ indicates that the back propagation of the gradients is stopped at this point, $Enc(\cdot)$ represents the mapping function of $E$, and $\alpha$ and $\beta$ are weighting parameters used to balance the importance of each item.

The networks of $E$ and $G$ are designed in a similar manner to other GANs [31, 18], except that $E$ has two output variables with respect to $\mu$ and $\sigma$. As shown in Algorithm 1, $E$ and $G$ are trained iteratively by updating $E$ using $L_E$ to distinguish the real data $X$ and generated samples, $X_r$ and $X_p$, and then updating $G$ using $L_G$ to generate samples that are increasingly similar to the real data; these steps are repeated until convergence.

## 4 Experiments

In this section, we conduct a set of experiments to evaluate the performance of the proposed method. We first give an introduction of the experimental implementations, and then discuss in detail the image quality, training stability and sample diversity of our method. Besides, we also investigate the learned manifold via interpolation in the latent space.

### 4.1 Implementations

**Dataset** We condider three data sets, namely CelebA [26] , CelebA-HQ [18] and LSUN BED-ROOM [40]. The CelebA dataset consists of 202,599 celebrity images with large variations in facial attributes. Following the standard protocol of CelebA, we use 162,770 images for training, 19,867 for validation and 19,962 for testing. The CelebA-HQ dataset is a high-quality version of CelebA that consists of 30,000 images at $1024 \times 1024$ resolution. The dataset is split into two sets: the first

29,000 images as the training set and the rest 1,000 images as the testing set. We take the testing set to evaluate the reconstruction quality. The LSUN BEDROOM is a subset of the Large-scale Scene Understanding (LSUN) dataset [40]. We adopt its whole training set of 3,033,042 images in our experiments.

**Network architecture** We design the inference and generator models of IntroVAE in a similar way to the discriminator and generator in PGGAN except of the use of residual blocks to accelerate the training convergence (see Appendix B for more details). Like other VAEs, the inference model has two output vectors, respectively representing the mean $\mu$ and the covariance $\sigma^2$ in Eq. (9). For the images at $1024 \times 1024$, the dimension of the the latent code is set to be 512 and the hyperparameters in Eq. (11) and Eq. (12) are set empirically to hold the training balance of the inference and generator models: $m = 90$, $\alpha = 0.25$ and $\beta = 0.0025$. For the images at $256 \times 256$, the latent dimension is 512, $m = 120$, $\alpha = 0.25$ and $\beta = 0.05$. For the images at $128 \times 128$, the latent dimension is 256, $m = 110$, $\alpha = 0.25$ and $\beta = 0.5$. The key is to hold the regularization term $L_{REG}$ in Eq. (11) and Eq. (12) below the margin value $m$ for most of the time. It is suggested to pre-train the model with $1 \sim 2$ epochs in the original VAEs form (i.e., $\alpha = 0$) to find the appropriate configuration of the hyper-parameters for different image sizes. More analyses and results for different hyper-parameters are provided in Appendix D.

As illustrated in Algorithm 1, the inference and generator models are trained iteratively using Adam algorithm [19] ($\beta_1 = 0.9$, $\beta_2 = 0.999$) with a batch size of 8 and a fixed learning rate of 0.0002. An additional illustration of the training flow is provided in Appendix C.

## 4.2 High quality image synthesis

As shown in Fig. 3, our method produces visually appealing high-resolution images of $1024 \times 1024$ resolution both in reconstruction and sampling. The images in Fig. 3(c) are the reconstruction results of the original images in Fig. 3(a) from the CelebA-HQ testing set. Due to the training principle of VAEs that injects random noise in the training phase, the reconstruction images cannot keep accurate pixel-wise similarity with the original images. In spite of this, our results preserve the most global topology information of the input images while achieve photographic high-quality in visual perception.

We also compare our sampling results against PGGAN [18], the state-of-the-art in synthesizing high-resolution images. As illustrated in Fig. 3(d), our method is able to synthesize high-resolution high-quality samples comparable with PGGAN, which are both distinguishable with the real images. While PGGAN is trained with symmetric generators and discriminators in a progressive multi-stage manner, our model is trained in a much simpler manner that iteratively trains a single inference model and a single generator in a single stage like the original GANs [13]. The results of our method demonstrate that it is possible to synthesize very high-resolution images by training directly with high-resolution images without decomposing the single task to multiple from-low-to-high resolution tasks. Additionally, we provide the visual quality results in LSUN BEDROOM in Fig. 4, which further demonstrate that our method is capable to synthesize high quality images that are comparable with PGGAN's. (More visual results on extra datasets are provided in Appendix F & G.)

## 4.3 Training stability and speed

Figure 5 illustrates the quality of the samples with regard to the loss functions of the reconstruction error $L_{AE}$ and the KL-divergences. It can be seen that the losses converge very fast to a stable stage in which their values fluctuate slightly around a balance line. As described in Theorem 1, the prediction $E(x)$ of the inference model reaches a constant $\gamma$ in $[0, m]$. This is consistent with the curves in Fig. 4, that when approximately converged, the KL-divergence of real images is around a constant value lower than $m$ while those of the reconstruction and sample images fluctuate around $m$. Besides, the image quality of the samples improves stably along with the training process.

We evaluate the training speed on CelebA images of various resolutions, i.e., $128 \times 128$, $256 \times 256$, $512 \times 512$ and $1024 \times 1024$. As illustrated in Tab. 1, The convergence time increases along with the resolution since the hardware limits the minibatch size for high-resolutions.

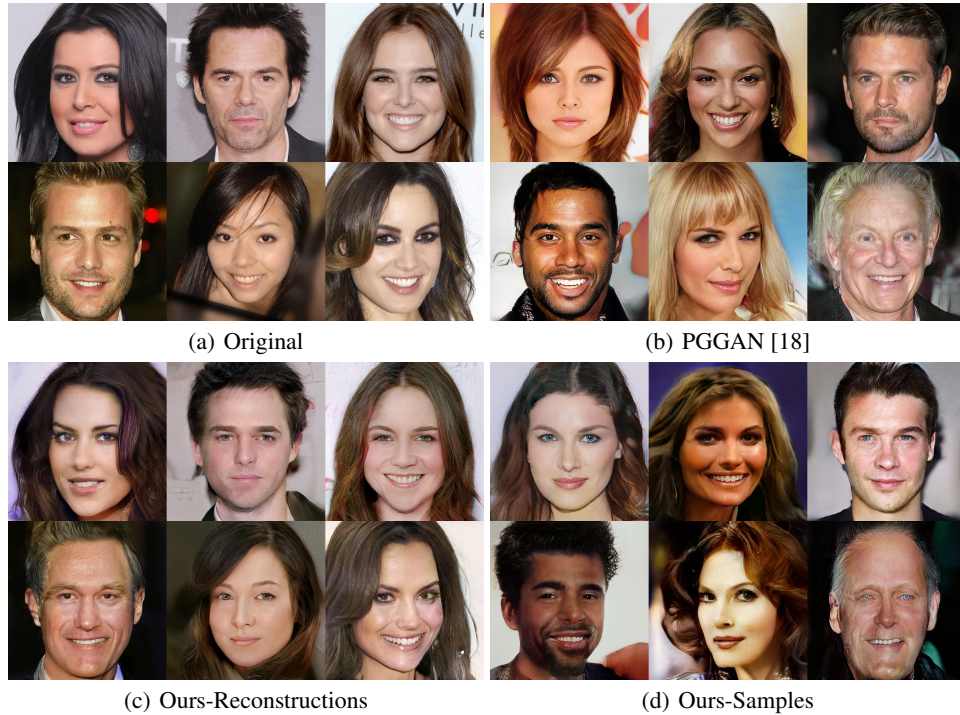

|            |            |
|:----------:|:----------:|
| (a) Original | (b) PGGAN [18] |
| (c) Ours-Reconstructions | (d) Ours-Samples |

Figure 3: Qualitative results of $1024 \times 1024$ images. (a) and (c) are the original and reconstruction images from the testing split, respectively. (b) and (d) are sample images of PGGAN (copied from the cited paper [18]) and our method, respectively. Best viewed by zooming in the electronic version.

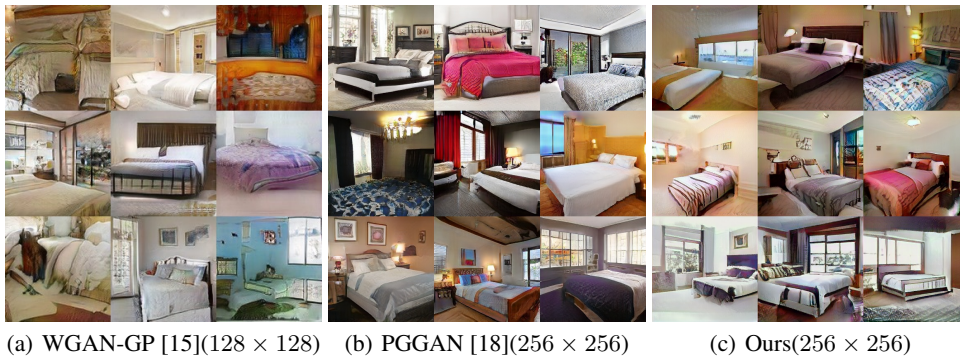

(a) WGAN-GP [15]($128 \times 128$)   (b) PGGAN [18]($256 \times 256$)   (c) Ours($256 \times 256$)

Figure 4: Qualitative comparison in LSUN BEDROOM. The images in (a) and (b) are copied from the cited papers [15, 18]

## 4.4 Diversity analysis

We take two metrics to evaluate the sample diversity of our method, namely **multi-scale structural similarity (MS-SSIM)** [30] and **Fréchet Inception Distance (FID)** [16]. The MS-SSIM measures the similarity of two images and FID measures the Fréchet distance of two distributions in feature space. For fair comparison with PGGAN, the MS-SSIM scores are computed among an average of

Table 1: Training speed w.r.t. the image resolutions.

| Resolution | $128 \times 128$ | $256 \times 256$ | $512 \times 512$ | $1024 \times 1024$ |
|:----------:|:----------------:|:----------------:|:----------------:|:------------------:|
| Minibatch  | 64               | 32               | 12               | 8                  |
| Time (days)| 0.5              | 1                | 7                | 21                 |

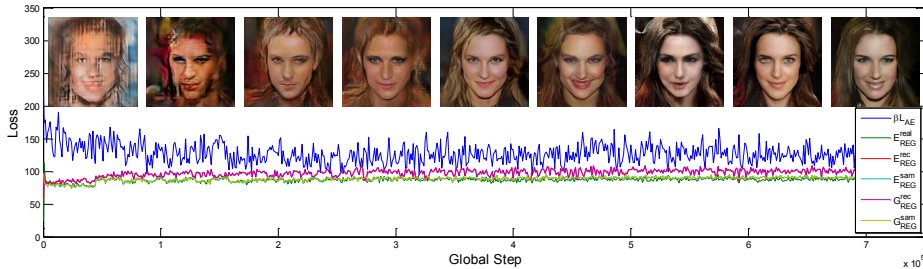

Figure 5: Illustration of the training process.

Table 2: Quantitative comparison with two metrics: MS-SSIM and FID.

| Method | MS-SSIM | | FID | |
|---|---|---|---|---|
| | CELEBA | LSUN BEDROOM | CELEBA-HQ | LSUN BEDROOM |
| WGAN-GP [15] | 0.2854 | 0.0587 | - | - |
| PGGAN [18] | 0.2828 | 0.0636 | 7.30 | **8.34** |
| Ours | **0.2719** | **0.0532** | **5.19** | 8.84 |

10K pairs of synthesize images at $128 \times 128$ for CelebA and LSUN BEDROOM, respectively. FID is computed from 50K images at $1024 \times 1024$ for CelebA-HQ and from 50K images at $256 \times 256$ for LSUN BEDROOM. As illustrated in Tab. 2, our method achieves comparable or better quantitative performance than PGGAN, which reflects the sample diversity to some degree. More visual results are provided in Appendix H to further demonstrate the diversity.

### 4.5 Latent manifold analysis

We conduct interpolations of real images in the latent space to estimate the manifold continuity. For a pair of real images, we first map them to latent codes $z$ using the inference model and then make linear interpolations between the codes. As illustrated in Fig. 6, our model demonstrates continuity in the latent space in interpolating from a male to a female or rotating a profile face. This manifold continuity verifies that the proposed model generalizes the image contents instead of simply memorizing them.

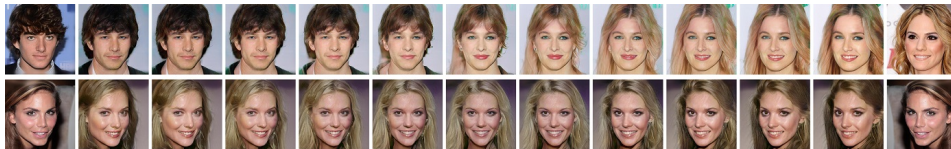

Figure 6: Interpolations of real images in the latent space. The leftmost and rightmost are real images in CelebA-HQ testing set and the images immediately next to them are their reconstructions via our model. The rest are the interpolations. The images are compressed to save space.

## 5 Conclusion

We have introduced introspective VAEs, a novel and simple approach to training VAEs for synthesizing high-resolution photographic images. The learning objective is to play a min-max game between the inference and generator models of VAEs. The inference model not only learns a nice latent manifold structure, but also acts as a discriminator to maximize the divergence of the approximate posterior with the prior for the generated data. Thus, the proposed IntroVAE has an introspection capability to self-estimate the quality of the generated images and improve itself accordingly. Compared to other state-of-the-art methods, the proposed model is simpler and more efficient with a single-stream network in a single stage, and it can synthesize high-resolution photographic images via a stable training process. Since our model has a standard VAE architecture, it may be easily extended to various VAEs-related tasks, such as conditional image synthesis.

**Acknowledgments**

This work is partially funded by the State Key Development Program (Grant No. 2016YFB1001001) and National Natural Science Foundation of China (Grant No. 61622310, 61427811).

## Footnotes

*It should be noted that we use $E$ to denote the inference model and $E(x)$ to denote the kl-divergence function for representation convenience.

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
