[Supplementary Material]

# A  Proof of theorem 1

Following the EBGAN [4], we give the proof as follows:

It is obvious that the sufficient conditions hold. So, we prove the necessary conditions. For the necessary condition $(a)$ $p_{G^*} = p_{data}$:

$(E^*, G^*)$ forms a saddle point that satisfies:

$$V(G^*, E^*) \le V(G^*, E) \qquad \forall E \tag{1}$$
$$U(G^*, E^*) \le U(G, E^*) \qquad \forall G \tag{2}$$

Firstly, $V(G^*, E)$ can be transformed as follows:

$$V(G^*, E) = \int_x p_{data}(x)E(x)\,\mathrm{d}x + \int_z p_z(z)\,[m - E(G^*(z))]^+\,\mathrm{d}z \tag{3}$$

$$= \int_x \left( p_{data}(x)E(x) + p_{G^*}(x)\,[m - E(x)]^+ \right)\mathrm{d}x \tag{4}$$

$$= \int_x \left( ay + b\,[m - y]^+ \right)\mathrm{d}x \tag{5}$$

where $a = p_{data}(x) \ge 0, y = E(x) \ge 0, b = p_{G^*}(x) \ge 0$. According to the analysis of $\varphi(y) = ay + b(m - y)^+$ in lemma A.1, which has been proved in [4],

**Lemma A.1** *Let $a, b \ge 0$, $\varphi(y) = ay + b\,[m - y]^+$. The minimum of $\varphi$ on $[0, +\infty)$ exists and is reached in $m$ if $a < b$, and it is reached in $0$ otherwise (the minimum may not be unique).*

$V(G^*, E)$ reaches its minimum when we replace $E^*(x)$ by these values.

$$V(G^*, E^*) = \int_x 1_{p_{data}(x)<p_{G^*}(x)} \left( p_{data}(x) \times 0 + p_{G^*}(x)\,[m - 0]^+ \right)\mathrm{d}x \tag{6}$$

$$+ \int_x 1_{p_{data}(x) \ge p_{G^*}(x)} \left( p_{data}(x) \times m + p_{G^*}(x)\,[m - m]^+ \right)\mathrm{d}x \tag{7}$$

$$= m\int_x 1_{p_{data}(x)<p_{G^*}(x)} p_{data}(x)\,\mathrm{d}x + m\int_x 1_{p_{data}(x) \ge p_{G^*}(x)} p_{G^*}(x)\,\mathrm{d}x \tag{8}$$

$$= m\int_x \left( 1_{p_{data}(x)<p_{G^*}(x)} p_{data}(x) + \left(1 - 1_{p_{data}(x)<p_{G^*}(x)}\right)p_{G^*}(x) \right)\mathrm{d}x \tag{9}$$

$$= m\int_x p_{G^*}(x)\,\mathrm{d}x + m\int_x 1_{p_{data}(x)<p_{G^*}(x)}(p_{data}(x) - p_{G^*}(x))\,\mathrm{d}x \tag{10}$$

$$= m + m\int_x 1_{p_{data}(x)<p_{G^*}(x)}(p_{data}(x) - p_{G^*}(x))\,\mathrm{d}x. \tag{11}$$

Since the second term in 11 $m\int_x 1_{p_{data}(x)<p_{G^*}(x)}(p_{data}(x) - p_{G^*}(x))\,\mathrm{d}x \le 0$, so $V(G^*, E^*) \le m$. By putting $p_{data}$ into the right side of equation 2, we get

$$\int_x p_{G^*}(x)E^*(x)\,\mathrm{d}x \le \int_x p_{data}(x)E^*(x)\,\mathrm{d}x. \tag{12}$$

$$\int_x p_{G^*}(x)E^*(x)\,\mathrm{d}x + \int_x p_{G^*}(x)[m - E^*(x)]^+\,\mathrm{d}x$$
$$\le \int_x p_{data}(x)E^*(x)\,\mathrm{d}x + \int_x p_{G^*}(x)[m - E^*(x)]^+\,\mathrm{d}x \tag{13}$$

$$\int_x p_{G^*}(x)E^*(x)\,\mathrm{d}x + \int_x p_{G^*}(x)[m - E^*(x)]^+\,\mathrm{d}x \le V(G^*, E^*) \tag{14}$$

According to lemma A.1, $E^*(x) \le m$ almost everywhere. So we get $m \le V(G^*, E^*)$.

Thus, $m \le V(G^*, E^*) \le m$ *i.e.* $V(G^*, E^*) = m$. Putting it into equation 11, $m + m\int_x 1_{p_{data}(x)<p_{G^*}(x)}(p_{data}(x) - p_{G^*}(x))\,\mathrm{d}x = m$, so we obtain $\int_x 1_{p_{data}(x)<p_{G^*}(x)}(p_{data}(x) - p_{G^*}(x))\,\mathrm{d}x = 0$. We can see that only if $p_G = p_{data}$ almost everywhere, the above equation is true.

Now for the necessary condition $(b)$ $E^*(x) = \gamma$ where $\gamma \in [0, m]$ is a constant. Following the proof by contradiction in [4]. Let us now assume that $E^*(x)$ is not constant almost everywhere and find a contradiction. If it is not, then there exists a constant $C$ and a set $\mathcal{S}$ of non-zero measure such that $\forall x \in \mathcal{S}, E^*(x) \leq C$ and $\forall x \notin \mathcal{S}, E^*(X) > C$. In addition we can choose $\mathcal{S}$ such that there exists a subset $\mathcal{S}' \subset \mathcal{S}$ of non-zero measure such that $p_{data}(x) > 0$ on $\mathcal{S}'$ (because of the assumption in the footnote). We can build a generator $G_0$ such that $p_{G_0}(x) \leq p_{data}(x)$ over $\mathcal{S}$ and $p_{G_0}(x) < p_{data}(x)$ over $\mathcal{S}'$. We compute

$$U(G^*, E^*) - U(G_0, E^*) = \int_x (p_{data} - p_{G_0})E^*(x)\,\mathrm{d}x \tag{15}$$

$$= \int_x (p_{data} - p_{G_0})(E^*(x) - C)\,\mathrm{d}x \tag{16}$$

$$= \int_{\mathcal{S}} (p_{data} - p_{G_0})(E^*(x) - C)\,\mathrm{d}x$$

$$+ \int_{\mathcal{R}^N \backslash \mathcal{S}} (p_{data} - p_{G_0})(E^*(x) - C)\,\mathrm{d}x \tag{17}$$

$$> 0 \tag{18}$$

which violates equation 2.

## B  Network Architecture

Tab. 1 is the network architecture for generating images of $1024 \times 1024$ resolution. We reduce the number of [Res-block + AvgPool] in the inference model and [Upsample + Res-block] in the generator for other smaller resolutions. In the experimental process we find that the residual block can accelerate the convergence for image synthesis, especially for resolutions larger than $256 \times 256$.

| Inference model | Act. | Output shape | Generator | Act. | Output shape |
|---|---|---|---|---|---|
| Input image | – | $3 \times 1024 \times 1024$ | Latent vector | – | $512 \times 1 \times 1$ |
| Conv | $5 \times 5, 16$ | $16 \times 1024 \times 1024$ | FC-8192 | ReLU | $8192 \times 1 \times 1$ |
| AvgPool | – | $16 \times 512 \times 512$ | Reshape | – | $512 \times 4 \times 4$ |
| Res-block | $\begin{bmatrix} 1 \times 1, & 32 \\ 3 \times 3, & 32 \\ 3 \times 3, & 32 \end{bmatrix}$ | $32 \times 512 \times 512$ | Res-block | $\begin{bmatrix} 3 \times 3, & 512 \\ 3 \times 3, & 512 \end{bmatrix}$ | $512 \times 4 \times 4$ |
| AvgPool | – | $32 \times 256 \times 256$ | Upsample | – | $512 \times 8 \times 8$ |
| Res-block | $\begin{bmatrix} 1 \times 1, & 64 \\ 3 \times 3, & 64 \\ 3 \times 3, & 64 \end{bmatrix}$ | $64 \times 256 \times 256$ | Res-block | $\begin{bmatrix} 3 \times 3, & 512 \\ 3 \times 3, & 512 \end{bmatrix}$ | $512 \times 8 \times 8$ |
| AvgPool | – | $64 \times 128 \times 128$ | Upsample | – | $512 \times 16 \times 16$ |
| Res-block | $\begin{bmatrix} 1 \times 1, & 128 \\ 3 \times 3, & 128 \\ 3 \times 3, & 128 \end{bmatrix}$ | $128 \times 128 \times 128$ | Res-block | $\begin{bmatrix} 3 \times 3, & 512 \\ 3 \times 3, & 512 \end{bmatrix}$ | $512 \times 16 \times 16$ |
| AvgPool | – | $128 \times 64 \times 64$ | Upsample | – | $512 \times 32 \times 32$ |
| Res-block | $\begin{bmatrix} 1 \times 1, & 256 \\ 3 \times 3, & 256 \\ 3 \times 3, & 256 \end{bmatrix}$ | $256 \times 64 \times 64$ | Res-block | $\begin{bmatrix} 1 \times 1, & 256 \\ 3 \times 3, & 256 \\ 3 \times 3, & 256 \end{bmatrix}$ | $256 \times 32 \times 32$ |
| AvgPool | – | $256 \times 32 \times 32$ | Upsample | – | $256 \times 64 \times 64$ |
| Res-block | $\begin{bmatrix} 1 \times 1, & 512 \\ 3 \times 3, & 512 \\ 3 \times 3, & 512 \end{bmatrix}$ | $512 \times 32 \times 32$ | Res-block | $\begin{bmatrix} 1 \times 1, & 128 \\ 3 \times 3, & 128 \\ 3 \times 3, & 128 \end{bmatrix}$ | $128 \times 64 \times 64$ |
| AvgPool | – | $512 \times 16 \times 16$ | Upsample | – | $128 \times 128 \times 128$ |
| Res-block | $\begin{bmatrix} 1 \times 1, & 512 \\ 3 \times 3, & 512 \\ 3 \times 3, & 512 \end{bmatrix}$ | $512 \times 16 \times 16$ | Res-block | $\begin{bmatrix} 1 \times 1, & 64 \\ 3 \times 3, & 64 \\ 3 \times 3, & 64 \end{bmatrix}$ | $64 \times 128 \times 128$ |
| AvgPool | – | $512 \times 8 \times 8$ | Upsample | – | $64 \times 256 \times 256$ |
| Res-block | $\begin{bmatrix} 3 \times 3, & 512 \\ 3 \times 3, & 512 \end{bmatrix}$ | $512 \times 8 \times 8$ | Res-block | $\begin{bmatrix} 1 \times 1, & 32 \\ 3 \times 3, & 32 \\ 3 \times 3, & 32 \end{bmatrix}$ | $32 \times 256 \times 256$ |
| AvgPool | – | $512 \times 4 \times 4$ | Upsample | – | $32 \times 512 \times 512$ |
| Res-block | $\begin{bmatrix} 3 \times 3, & 512 \\ 3 \times 3, & 512 \end{bmatrix}$ | $512 \times 4 \times 4$ | Res-block | $\begin{bmatrix} 1 \times 1, & 16 \\ 3 \times 3, & 16 \\ 3 \times 3, & 16 \end{bmatrix}$ | $16 \times 512 \times 512$ |
| Reshape | – | $8192 \times 1 \times 1$ | Upsample | – | $16 \times 1024 \times 1024$ |
| FC-1024 | – | $1024 \times 1 \times 1$ | Res-block | $\begin{bmatrix} 3 \times 3, & 16 \\ 3 \times 3, & 16 \end{bmatrix}$ | $16 \times 1024 \times 1024$ |
| Split | – | $512, 512$ | Conv | $5 \times 5, 3$ | $3 \times 1024 \times 1024$ |

Table 1: Network architecture for generating $1024 \times 1024$ images.

## C Illustration of training flow

As illustrated in Fig. 1, we train the inference model and generator iteratively that an extra pass through the inference model is necessary after images are generated or reconstructed. As in the algorithm 1, we use $ng(\cdot)$ to stop the gradients of $L_{adv}^E$ (Line (8) and (9) in the Algorithm 1) propagating back to the generator in the first pass. For other choices, such as no $ng(\cdot)$ or updating the generator first, it also works with one forward pass through the inference model. The current choice is for realization convenience.

(a) Updating the inference model.

(b) Updating the generator.

Figure 1: The training flow of Algorithm 1. The solid and dash lines illustrate the forward and backward passes of the proposed model, respectively. The inference model and generator are updated iteratively.

## D Discussion of hyper-parameters

We conduct experiments on the images of $256 \times 256$ in CELEBA-HQ and find the training stability is not very sensitive to the hyper-parameters in some degree though they indeed have influences on the sample and reconstruction quality. $\alpha$ is better to be $0.1 \sim 0.5$ and larger or smaller may decelerate the convergence speed. As illustrated in Fig. 2, when $\alpha$ is fixed, larger $\beta$ always improves the reconstruction quality but may influence the sample diversity. The margin $m$ should be selected according to the value of $\beta$ because larger $\beta$ needs larger $m$ to balance the adversarial training. Pre-training the model following the original VAEs (i.e., $\alpha = 0$) is suggested to find the most appropriate value of $m$ responding to a specific $\beta$. $m$ can be selected to be a little larger than the training kl-divergence value of VAEs.

Figure 2: Results of different hyper-parameters where $\alpha$ is fixed to be 0.25. For each setting, the first column of images are the reconstructions and the second are the samples. We use RMSE (smaller is better) to describe the reconstruction quality and MS-SSIM (smaller is better) to demonstrate the sample diversity.

# E  Nearest neighbors for the generated images

Fig. 3 shows the nearest neighbors from the training data for the generated images (the first row in Fig. 3). We find the nearest neighbors using two distance measures: the second row of images in Fig. 3 are the results based on $L_1$ distance in pixel space; the bottom row of images are the results based on cosine distance in feature space. The high-level features are extracted using a pretrained face recognition network, i.e. LightCNN [2].

Figure 3:  Nearest neighbors for the generated images.

# F  Qualitative comparison on LSUN CHURCHOUTDOOR

(a) PGGAN

(b) Ours

Figure 4:  Qualitative comparison on LSUN CHURCHOUTDOOR [3]. The images in (a) are copied from the cited papers [1]

# G    Qualitative comparison on DOG images

(a) PGGAN

(b) Ours

Figure 5:  Qualitative comparison on DOG images. Our model is trained with $256 \times 256$ dog images from the ImageNet database. The images in (a) are copied from the cited papers [1].

## H  Additional $1024 \times 1024$ images

Figure 6:  Additional results of $1024 \times 1024$ images.

Figure 7: Additional results of $1024 \times 1024$ images.