[Reviews · NeurIPS 2018]

Reviewer 1



Update: I raised my score by two points because the rebuttal and reviews/comments revealed more differences that I originally noticed with respect to the AGE work, in particular in terms of the use of the KL divergence as a discriminator per example, and because the authors promised to discuss the connection to AGE and potentially expand the experimental section. I remain concerned that the resulting model is not a variational auto-encoder anymore despite the naming of the model (but rather closer to a GAN where the discriminator is based on the KL divergence), and about the experimental section, which reveals that the method works well, but does not provide a rich analysis for the proposed improvements. --- The paper proposes an approach for adversarial learning in the context of variational auto-encoders. Rather than using a separate discriminator network, the work proposes a learning objective which encourages the encoder to discriminate between real data and generated data: it guides the approximate posterior to be close to the prior in the real data case and far from the prior otherwise. The approach is illustrated on the task of synthesizing high-resolution images, trained on the CelebA-HQ dataset. First, high-quality image generation remains an important area of research, and as a result, the paper's topic is relevant to the community. Furthermore, the results are very good, comparable to state of the art, despite not using a discriminator network. However, I believe the paper does not position itself strongly with respect to the cited work, AGE [34]. The authors highlight the following differences: "design of inference models, the training objects, and the divergence computations," but without detail. To me, the approaches seem quite similar: in both cases, the objective relies on decreasing the divergence between the posterior and the prior for real data, and increasing it for generated data; in both cases the chosen divergence measure is a KL divergence, and the reference (prior) distribution is a Gaussian; etc.. There are slight differences such as the use of an upper bound (m) on the divergence between the generated data posterior and the prior, but a discussion of why they might be needed is missing. Could you discuss this in your rebuttal? A major point made in the write-up is that previous methods based on encoder-decoder architectures (without discriminators), including AGE, have obtained weaker results and that the submission is the first to generate high-resolution images. This result is promising, but I think it should come with an understanding of the differences to the previous works, highlighting which modifications have caused the improvement. Otherwise, the improvement could be attributed to better hyper-parameter tuning. Instead, an empirical comparison with the cited work, AGE, is not provided at all. Other comments: * In lines 127-132, the use of E(x) to denote the KL divergence loss, as well as referring to E as the inference (encoder) model creates confusion. Same with E* later. Summary: The current work presents good image synthesis results for an adversarially trained VAE architecture with no discriminator. However, the approach appears very similar to previous work, making it less relevant for the NIPS community.

Reviewer 2



The paper combines variational auto-encoder training with adversarial training in a novel way. Rather then having a separate discriminator, the KL term of VAE is used as the discriminator (as well as in the same way as in VAE). The goal of the KL term as discriminator is to assign high likelihood (low KL) to input samples and low likelihood to both samples and reconstructions. The generator tries to generate samples that have high likelihood under (low KL). The resulting training seems the be more stable and 1024x1024 samples of faces are produced without stage-wise training. The method is nice and natural and the paper is done well (see some comments below though). However there should be (at least) two important improvements for the paper: 1. Show how sensitive the training is to the hyper parameters (how well they need to be tuned to get good results and what the results look like for different values), 2. Train on rich dataset such as ImageNet where training is not all about overfitting. Detailed comments: - It would be good to show generated sample and performance measure for different hyperparametners (e.g. as a 2d grid). - Why not training on ImageNet? - 182-183: It is a little confusing discussion regarding two types of samples - reconstruction and samples (x_r and x_p). One might think that we discrimintate these two from each other, which might also be reasonable given that reconstruction can be close to the input. Line 195 talks about x vs x_r, x_p but it would be good to say it earlier: We discriminate real sample FROM both the model samples and reconstructions. - Formula (11) has alpha in front of both x_r and x_p but the Algorithm has it only in front of one of them. Which on is correct? - The reconstructions in Figure 3 are quite far from the originals - is it because beta is so small? What happens when it is larger (as mentioned before, it would be good to see what happens for different hyper parameter settings - not only the quality actually but also their property). - Is there a reason why in Algorithm 1, the there is beta for the L_AE but there is no beta for the same gradient going into the encoder? - 215: alpha -> \alpha

Reviewer 3



The work presents a version of VAEGAN (a VAE whose decoder is replaced with a GAN) with interesting contributions in the training methodology and in the quality of results generated. The formulation is a development on ideas from the EBGAN (Energy based GAN) and an earlier work (Autoencoding pixels beyond a learned similarity metric). General comments 1) This is a VAE/GAN in which the KL divergence term of the VAE is adapted to setup the GAN minmax game, wherein the first player tries to minimize the KL divergence between real samples and prior, while maximizing it for the generated samples. The formulation is developed based on EBGAN, with the discriminator in the EBGAN now being replaced by the KL divergence term of the VAE. To this, a reconstruction term is added to complete the VAE setup [regularizer + reconstruction terms] 2) A training methodology in which one does not need a separate discriminator to train the GAN. The training procedure itself is similar to the older VAE/GAN paper "Autoencoding Pixels beyond a learned similarity metric" [ref 21 in the current paper]. The same machinery of [ref 21] seems to be adopted to add additional signal to the system by decoupling the encoder and decoder. 3) The results are extremely impressive. The authors present comparison with the highly regarded "PCGAN" as reference. In addition to the above, the paper is a good survey of existing flavors of GANs and VAEs. Questions and clarifications: One aspect that wasn't very clear to me was why one needed an extra pass through the encoder after images were generated )reconstructed) from G to use in the GAN, instead of using mu and sigma generated from the first pass. Minor comment: It would help to have a flow diagram describing the encoder+generator setups although, they are presumably the same as in [ref 21]. Minor comment: the formula (12) is inconsistent with formula (11) in that L_{AE} should take in two inputs (x,x_r) in (11) whereas it only has one input in (12) L_AE(x). One of the main points of the paper seems to be that the training methodology results in better samples. However, the paper does not discuss why this should be so, in the sense that while the samples produced do indeed seem to be excellent, the circumstance seems to be arbitrary. Perhaps we should have a few studies (empirical or theoretically motivated) explaining why we get good samples, and what the practitioner should do to get them by way of tuning. Specific comments on review criteria: 1) Quality: Good. The paper is well written, easy to follow and generates excellent results. 2) Clarity: Very good. 3) Originality: Highly original. This development of a GAN without a separate discriminator is novel. However, the AGE paper has already demonstrated that this can be done. 4) Significance: Extremely significant. The reviewer sees this method being of great practical use in VAE architectures in general. Pertinent practical examples are domain adaptation problems where it is necessary to polish samples produced by VAE to be of practical use.